# Maximizing Confidence Alone Improves Reasoning

**Mihir Prabhudesai\*, Lili Chen\*, Alex Ippoliti\***
*mprabhud,lilic,aippolit@andrew.cmu.edu*
*\*Equal contribution*
*Carnegie Mellon University*

**Katerina Fragkiadaki, Hao Liu, Deepak Pathak**
*Carnegie Mellon University*

**Reviewed on OpenReview:** *https://openreview.net/forum?id=gInznr8EsQ*

## Abstract

Reinforcement learning (RL) has enabled machine learning models to achieve significant advances in many fields. Most recently, RL has empowered frontier language models to solve challenging math, science, and coding problems. However, central to any RL algorithm is the reward function, and reward engineering is a notoriously difficult problem in any domain. In this paper, we propose **RENT: R**einforcement Learning via **Ent**ropy Minimization – a fully unsupervised RL method that requires no external reward or ground-truth answers, and instead uses the model's entropy of its underlying distribution as an intrinsic reward. We find that by reinforcing the chains of thought that yield high model confidence on its generated answers, the model improves its reasoning ability. In our experiments, we showcase these improvements on an extensive suite of commonly-used reasoning benchmarks, including GSM8K, MATH500, AMC, AIME, and GPQA, and models of varying sizes from the Qwen, Mistral, and Llama families. The generality of our unsupervised learning method lends itself to applicability in a wide range of domains where external supervision is unavailable.

## 1 Introduction

Imagine you're taking an exam. Once it begins, no new information is available and no external help can be sought. With only your own reasoning to rely on, how do you tackle a difficult problem? You might make an initial attempt, assess your confidence in the answer, and revise your reasoning until you feel sufficiently certain. Of course, confidence is not a guarantee of correctness – but in the absence of feedback, it is often the only intrinsic signal we have to guide further thought. In such settings, humans tend to optimize for confidence, or equivalently, to reduce uncertainty. In machine learning, uncertainty is commonly quantified via entropy – a measure of how peaked or diffuse a probability distribution is. Language models output distributions over tokens, and the entropy of these distributions reflects the model's confidence: lower entropy implies more confident predictions. Yet despite the growing use of language models in reasoning tasks, current approaches to improvement still rely heavily on external supervision, rewarding correctness with respect to ground-truth labels Guo et al. (2025); Shao et al. (2024). This dependence is often impractical, particularly in real-world or open-ended scenarios where supervision is scarce or unavailable.

To address this, we propose **RENT: R**einforcement Learning via **Ent**ropy Minimization – a fully unsupervised reinforcement learning method that improves reasoning performance by using the model's own confidence as a reward. Specifically, we define the reward as the negative entropy of the model's predicted token distributions. This signal is dense, general, and easy to compute, requiring no ground-truth answers. Importantly, not all parts of the response contribute equally to final performance. Through empirical analysis, we find that minimizing entropy over tokens near the end of the reasoning chain, especially those corresponding to the final answer, correlates most strongly with improved accuracy. In contrast, early to-

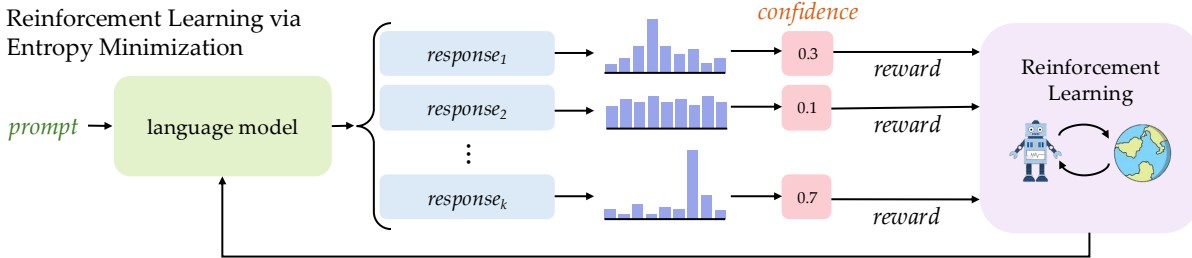

Figure 1: Overview of RENT: Reinforcement Learning via Entropy Minimization. For each response, we use the model's underlying confidence (negative entropy) as a reward for reinforcement learning. This enables the model to learn without any external reward or ground-truth answers.

kens in the response show little correlation. This suggests that as the model approaches its final answer, it increasingly relies on its confidence to guide reasoning, so encouraging confidence in these final steps is key.

Concretely, we frame the problem through the lens of test-time training (TTT), where a model continues to adapt even after deployment, using only information available at inference. Unlike conventional training, which relies on labeled data and offline optimization, TTT updates model parameters or behavior directly from the input distributions observed during testing. We demonstrate RENT's effectiveness across diverse reasoning benchmarks, including GSM8K (Cobbe et al., 2021), MATH500 (Hendrycks et al., 2021; Lightman et al., 2023), AMC and AIME (Li et al., 2024), and GPQA (Rein et al., 2024). Our method scales across model families (Qwen, Mistral, and Llama) and sizes and consistently improves performance.

## 2 Related Work

### 2.1 Reinforcement Learning for Reasoning

Initially, reinforcement learning (RL) for language models was mostly used for learning from human preferences (Christiano et al., 2017) and, traditionally, RL optimization was done with algorithms such as PPO (Schulman et al., 2017). With the capabilities of language models continuing to improve, researchers have begun to explore the possibility of using RL to improve the performance of language models on reasoning tasks such as math (Cobbe et al., 2021; Hendrycks et al., 2021; Li et al., 2024), science (He et al., 2024; Rein et al., 2024), or coding (Li et al., 2022; Chaudhary, 2023) problems. In these settings, the model is prompted to generate a chain-of-thought (Wei et al., 2022) and final answer, and receives a reward based on how closely its final answer matches the ground truth. These efforts present RL as an alternative to search-based approaches to chain-of-thought reasoning such as Tree of Thoughts (Yao et al., 2023) and Graph of Thoughts (Besta et al., 2024). Related lines of work include training a reward model to give feedback for every step in the chain of thought, and training RL models to encourage self-correcting behaviors in language models. Examples of RL methods in this space include Zelikman et al. (2022); Singh et al. (2023); Kumar et al. (2024); Qu et al. (2024); Uesato et al. (2022); Lightman et al. (2023); Wang et al. (2023). At scale, DeepSeek (Guo et al., 2025; Shao et al., 2024) proposed an open-source model that showed OpenAI o1 (Jaech et al., 2024)-level reasoning by performing RL in this manner, using GRPO (Shao et al., 2024).

### 2.2 Confidence and Calibration

Confidence measures quantify how certain a model is that its generated output is correct (Yoon et al., 2025; Spiess et al., 2024). In order to evaluate the confidence of machine learning models, it is necessary also to discuss *calibration* (Kalai & Vempala, 2024; Virk et al., 2024) - i.e., how aligned those confidences are with actual correctness. As language models are increasingly trusted to make important decisions, providing users with a reliable confidence measure would be useful in many situations (Geng et al., 2023; Manakul et al., 2023; Varshney et al., 2023; Hou et al., 2023; Jiang et al., 2023; Han et al., 2024; Spiess et al., 2024). As such, researchers have developed various confidence metrics for deep learning models and studied the extent

to which they are calibrated. These include both methods that assume access to the model's weights (Gupta et al., 2024; Kadavath et al., 2022; Xu et al., 2024; Spiess et al., 2024) and methods that estimate confidence via prompting alone (Xie et al., 2024; Geng et al., 2023; Tian et al., 2023; Xiong et al., 2023; Yang et al., 2024). In our paper, we use the model's confidence to iteratively improve its own performance via RL.

### 2.3 Test-Time Adaptation

Test-time adaptation is where a model is updated using data from the test distribution, without access to ground-truth labels. The goal is to improve performance in scenarios where there is a distribution shift between training and testing environments. Methods for adapting without labels include normalization techniques that recalibrate feature statistics at test time (Quiñonero-Candela et al., 2008; Sun et al., 2017; Maria Carlucci et al., 2017; Schneider et al., 2020; Nado et al., 2020). The most relevant work to ours is Tent (Wang et al., 2020), which performs entropy minimization on model predictions during test time. This approach assumes that predictions on test data should be low in entropy if the model is well-adapted to the new distribution. Tent builds on earlier work that uses entropy minimization as a regularization strategy in semi-supervised learning (Grandvalet & Bengio, 2004; Lee et al., 2013; Berthelot et al., 2019) and domain adaptation contexts (Maria Carlucci et al., 2017; Shu et al., 2018; Saito et al., 2019), where encouraging confident predictions has proven effective for improving generalization. Recently, TTRL (Zuo et al., 2025) proposed test-time RL using majority voting as a reward. Compared to entropy, majority voting is a sparse reward and less general; for example, it cannot be applied to long-form free-response questions.

### 2.4 Unsupervised Reinforcement Learning

Unsupervised RL trains agents using intrinsic rewards like novelty, entropy, or mutual information, enabling skill acquisition without extrinsic feedback. Prior methods include ICM and RND for prediction error Pathak et al. (2017); Burda et al. (2018), APT Liu & Abbeel (2021) and ProtoRL Yarats et al. (2021) for entropy maximization, and DIAYN, APS, and SMM for skill discovery via mutual information Eysenbach et al. (2018); Liu & Abbeel (2021); Kim et al. (2023). An interesting observation is that while exploration methods primarily maximize entropy, we instead minimize it by reinforcing high-confidence outputs, and find that for language models, this leads to better reasoning performance without any external supervision.

## 3 Method

### 3.1 Reinforcement Learning for Language Models

The goal of reinforcement learning (RL) is to train a policy which generates actions that maximize the cumulative expected reward. In the context of modern language models, the policy $\pi$ is a language model and the actions $y_{\text{pred}}$ are sampled from the distribution over a discrete vocabulary. The task is formulated as a one-step RL problem in which the model generates $y_{\text{pred}} = \pi(x)$, where $x$ is sampled from the dataset $\mathcal{D} = \{(x, y_{\text{target}})\}$, and receives some reward for the generation. Typically, the ground-truth answer $y_{\text{target}}$ is used to give the model a reward $r = \mathcal{R}(y_{\text{target}}, y_{\text{pred}})$. One reward function which is currently used is simple string matching, where $\mathcal{R}(y_{\text{target}}, y_{\text{pred}}) = \mathbb{1}\{y_{\text{target}} = y_{\text{pred}}\}$. Our work focuses on instead doing *unsupervised* reinforcement learning, which does not require external supervision for the reward. Specifically, $y_{\text{target}}$ is not used in the reward $r = \mathcal{R}(y_{\text{pred}})$ and we do not assume access to this at any point in training. Crucially, we frame this entirely as a test-time training (TTT) procedure; the policy $\pi$ is updated directly on the evaluation target distributions at test-time to adapt its outputs without any access to ground-truth answers, rather than aiming to serve as a generalizable, multi-task offline pretraining objective.

### 3.2 Group Relative Policy Optimization (GRPO)

To optimize the policy, we adopt Group Relative Policy Optimization (GRPO; Shao et al., 2024), a critic-free variant of PPO that measures *relative* performance within a group of rollouts for the same input, rather than against a separate baseline policy.

For each training query $x$, we sample a group of $G$ candidate responses $\{y^{(1)}, \ldots, y^{(G)}\} \sim \pi_{\text{old}}(\cdot \mid x)$, and compute their rewards $r^{(i)} = \mathcal{R}(x, y^{(i)})$. GRPO normalizes these rewards within the group to form a group-relative advantage:

$$\mu_x = \frac{1}{G} \sum_{i=1}^{G} r^{(i)}, \qquad \sigma_x = \sqrt{\frac{1}{G} \sum_{i=1}^{G} \left(r^{(i)} - \mu_x\right)^2}, \qquad A^{(i)} = \frac{r^{(i)} - \mu_x}{\sigma_x + \varepsilon},$$

where $\mu_x$ and $\sigma_x$ denote the mean and standard deviation of rewards in the group, and $\varepsilon$ is a small constant for numerical stability. Intuitively, $A^{(i)}$ quantifies how much better or worse a response is relative to its peers for the same prompt.

Using these group-relative advantages, GRPO applies a PPO-style clipped objective to update the current policy $\pi_\theta$ away from $\pi_{\text{old}}$:

$$\mathcal{L}_{\text{GRPO}}(\theta) = \mathbb{E}_{x, \{y^{(i)}\} \sim \pi_{\text{old}}} \left[ \frac{1}{G} \sum_{i=1}^{G} \min\left( \rho_\theta^{(i)} A^{(i)}, \ \text{clip}\left(\rho_\theta^{(i)}, 1 - \epsilon, 1 + \epsilon\right) A^{(i)} \right) \right],$$

where $\rho_\theta^{(i)} = \frac{\pi_\theta(y^{(i)}|x)}{\pi_{\text{old}}(y^{(i)}|x)}$ is the importance ratio and $\epsilon$ is the PPO clipping parameter. For more details, we refer the reader to Shao et al. (2024).

### 3.3 Entropy Reward

For a given prompt $x$, the model generates a response $y_{\text{pred}} = y_{\text{pred},1}, \cdots, y_{\text{pred},T} = \pi(x)$, where $T$ is the number of tokens in the response. At each token $t \in \{1, \ldots, T\}$, the model outputs a probability distribution $p_t$ over the vocabulary $\mathcal{V}$, i.e., $p_t(v) = P(y_t = v \mid x, y_{<t})$. The entropy of this distribution measures the model's uncertainty in predicting the next token and is given by:

$$H(p_t) = - \sum_{v \in \mathcal{V}} p_t(v) \log p_t(v)$$

To compute the total entropy of the response, we average the entropies across all tokens. The total entropy $H(\pi(x))$ provides a measure of the overall uncertainty in the model's response. Higher entropy indicates greater uncertainty or more diverse token predictions, while lower entropy suggests more confident and peaked distributions at each token. We use the negative entropy of the predicted token distribution as a reward signal:

$$\mathcal{R}(y_{\text{pred}}) = -H(\pi(x)) = \frac{1}{T} \sum_{t=1}^{T} \sum_{v \in \mathcal{V}} p_t(v) \log p_t(v)$$

This reward encourages the model to produce more confident and peaked distributions over the vocabulary, effectively promoting lower uncertainty in its predictions. Within the RL framework, the learning objective becomes maximizing the expected reward over the data distribution:

$$\max_{\pi} \mathbb{E}_{x \sim \mathcal{D}} \left[ \mathbb{E}_{y_{\text{pred}} \sim \pi(x)} \left[ \mathcal{R}(y_{\text{pred}}) \right] \right]$$

By optimizing this objective, the model learns to generate responses with lower entropy without relying on external supervision or labeled target responses.

## 4 Experiments

### 4.1 Experimental Setup

**Benchmarks.** We train a model with reinforcement learning on each dataset independently. We conduct our experiments on the following commonly-used benchmarks for evaluating the reasoning capabilities of large language models:

- GSM8K (Cobbe et al., 2021): GSM8K contains 8792 grade-school math word problems. The train set contains roughly 7473 problems and the test set contains roughly 1319 problems.

- MATH500 (Hendrycks et al., 2021; Lightman et al., 2023): MATH (Hendrycks et al., 2021) is a dataset containing competition math problems spanning seven categories. It contains 12500 problems, of which 7500 are used for training and 5000 are used for testing. MATH500 (Lightman et al., 2023) is a subset of the MATH test set created by sampling uniformly at random from the test set.

- AMC (Li et al., 2024): The American Mathematics Competitions (AMCs) are competitions given to high school students. The specific dataset we use is comprised of 83 problems from the 2022 and 2023 AMC12 exams. Although the original problems are in multiple-choice format, the dataset presents modified versions of the problem which expect an integer solution.

- AIME24 (Li et al., 2024): The American Invitational Mathematics Examination (AIME) is a prestigious high school mathematics competition. It consists of 15 questions in 3 hours and is given to top-scoring students on the AMC exam. Each year, there are two versions of the exam which consist of distinct questions. We train on the 30 problems from both versions of the 2024 exam.

- GPQA (Rein et al., 2024): GPQA is a dataset of 448 multiple-choice problems in biology, physics, and chemistry at the PhD level. They are intended to be "Google-proof" in the sense that they require advanced reasoning skills.

Since we are interested in test-time adaptation, and we do not assume access to the ground-truth answer, we use the same dataset for training and evaluation. Additionally, some benchmarks do not have standardized train sets. The exception is GSM8K, where we use the standard train and test sets; this shows that generalization does occur and RENT is not merely overfitting to the test set.

**Models.** We conduct experiments on a wide range of models from different model families and with varying parameter counts: Mistral-7B-Instruct-v0.3, Llama3.1-8B-Instruct, Qwen2.5-1.5B-Instruct, Qwen2.5-Math-1.5B-Instruct, Qwen2.5-7B-Instruct, and Qwen2.5-Math-7B-Instruct.

**Implementation details.** For the RL optimization we use GRPO (Shao et al., 2024) with a learning rate of $1 \times 10^{-6}$ and the Adam optimizer. The batch sizes and sampling hyperparameters may vary among models and datasets. We provide a full list of hyperparameters in the Appendix.

### 4.2 Main Results

Figure 2 shows the performance of models before and after entropy minimization on GSM8K, MATH500, AMC, AIME24, and GPQA. We report standard deviations over 5, 5, 32, 64, and 10 samples, respectively. Note that all models are Instruct models (e.g., Qwen2.5-1.5B refers to Qwen2.5-1.5B-Instruct). Across model families, model sizes, and benchmarks, entropy minimization allows large language models to improve their reasoning skills, without any external supervision. On the Math models such as Qwen2.5-Math-1.5B and Qwen2.5-Math-7B, the base model often struggles at following instructions and therefore the initial score is zero or near zero, and therefore the boost from entropy minimization is quite large. On models that are already proficient at instruction following, we can still see strong performance improvements from entropy minimization. Given the potential pitfall of overconfidence in language models, we performed extensive experimentation to ensure empirically that entropy minimization is a robust and generalizable reward function across datasets and models.

### 4.3 Is It Just Formatting?

It is a well-known issue with reasoning benchmarks that language models can lose points simply because they do not know how to put their answers in the right format. For example, MATH500 expects final answers to be placed in "boxed". A nontrivial amount of engineering effort has gone into both designing prompts that encourage correct formatting and implementing parsers that effectively extract answers from language model responses, in attempts to mitigate this issue. Therefore, one might wonder if, instead of learning to perform complex reasoning, RENT merely encourages the model to put its answers in the right format. Table 1 shows

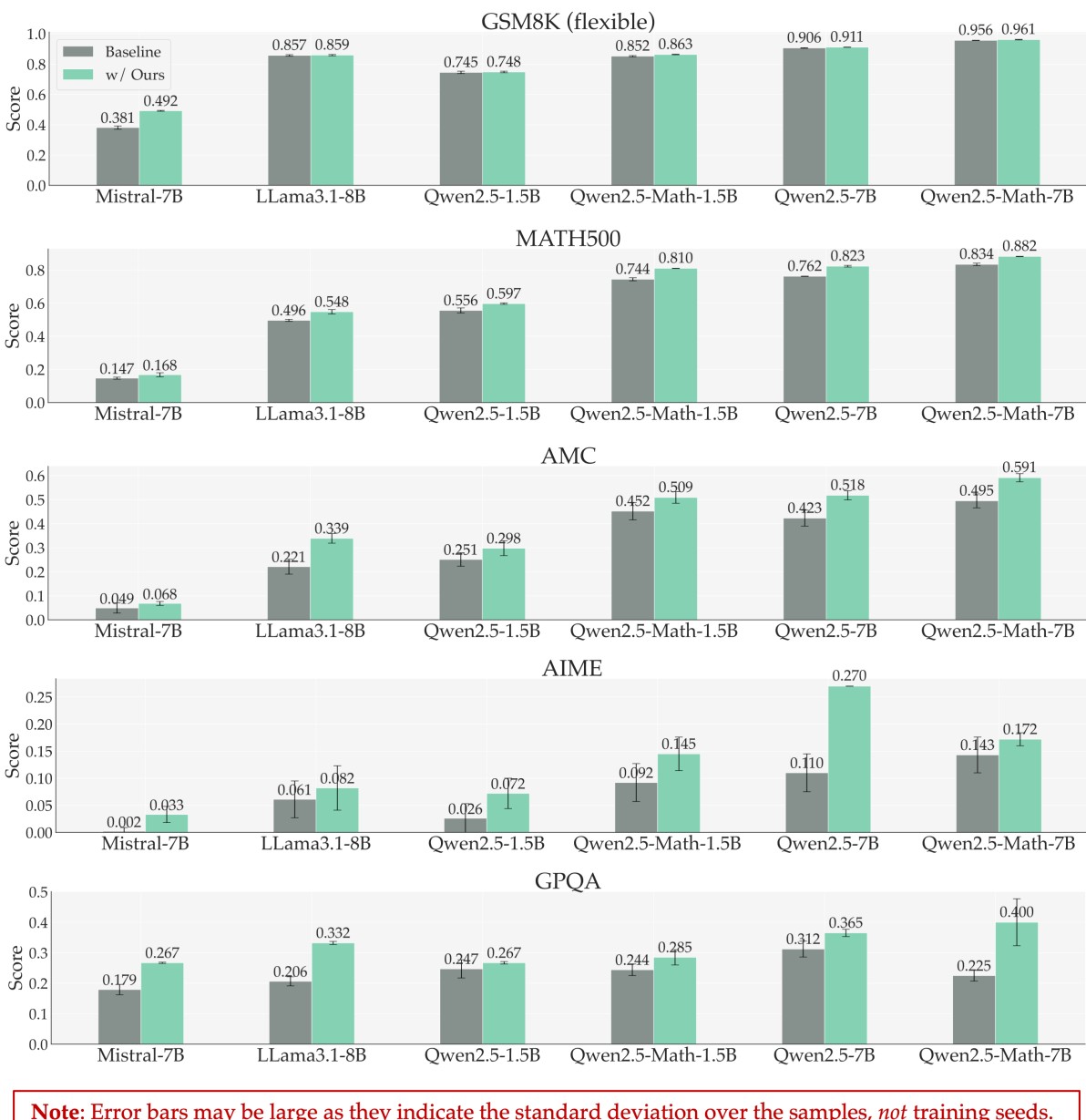

**Note**: Error bars may be large as they indicate the standard deviation over the samples, *not* training seeds.

Figure 2: Performance on GSM8K, MATH500, AMC, AIME, and GPQA. The standard deviations reported are over 5, 5, 32, 64, and 10 samples, respectively. Across benchmarks and models, we find that entropy minimization alone is an effective reward for improving the reasoning ability of language models. All models are Instruct models; the "Instruct" is omitted for brevity.

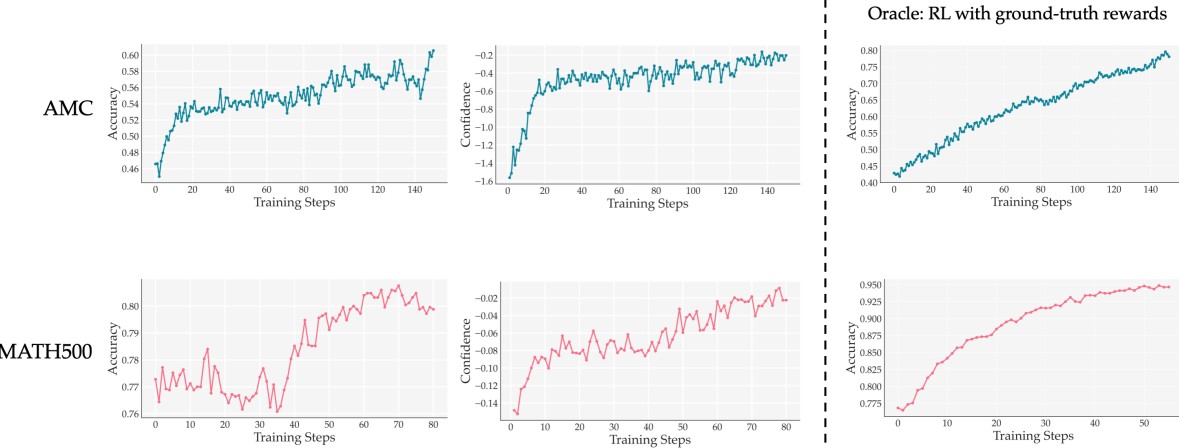

Figure 3: Accuracy and confidence over the course of training. The trends indicate that accuracy and confidence are indeed highly correlated and therefore it is natural to use confidence as a reward.

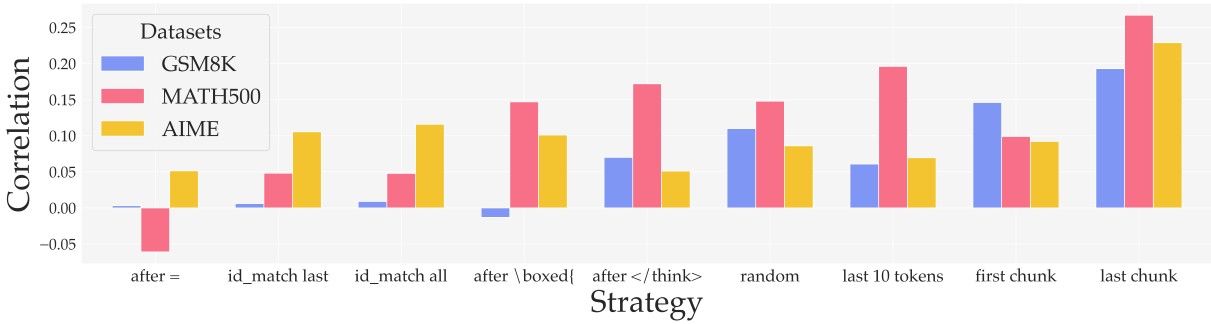

Figure 4: Evaluation (by computing correlation between accuracy and confidence) of various strategies for selecting which tokens to minimize the entropy over. We find the highest correlation between accuracy and confidence in the last few tokens of the response.

that this is not the case. Models trained with the RENT reward outperform only using a format reward, which simply assigns a binary reward based on whether the correct format is followed in the response. In some cases, the performance of our method is similar to (or even slightly worse than) just using format reward, but of course it is expected that unsupervised RL methods might not always lead to significant improvements. For example, if the benchmark is extremely easy and the model only needs to learn the right format to achieve near-perfect scores, RENT would not outperform format reward. Or, if the benchmark is so hard that it is beyond the model's capabilities altogether, neither method would perform well. However, across datasets and model sizes, we find a consistent improvement over using the format reward and this assures us that the model is actually learning to think through difficult problems and improve its ability to reason.

## 4.4 Correlation Between Entropy and Accuracy

Figure 3 shows the accuracy and confidence throughout training Qwen2.5-Math-7B and Qwen2.5-7B-Instruct on the AMC and MATH500 datasets respectively. Critically, as the model improves its confidence via RENT, the accuracy of the model improves as well. This demonstrates the significant correlation between answer confidence and answer accuracy, supporting our initial hypothesis.

Table 1: Comparison to RL with a format reward. The best result on each benchmark is indicated in bold. RENT generally outperforms only using a format reward.

|  | GSM8K | MATH500 | AMC | AIME | GPQA |
|---|---|---|---|---|---|
| *Mistral-7B-Instruct-v0.3* | 0.381 | 0.147 | 0.049 | 0.002 | 0.179 |
| w/ Format reward only | 0.393 | 0.150 | 0.051 | 0.015 | 0.240 |
| w/ RENT (Ours) | **0.492** | **0.168** | **0.068** | **0.033** | **0.267** |
| *LLama3.1-8B-Instruct* | 0.857 | 0.496 | 0.221 | 0.061 | 0.206 |
| w/ Format reward only | **0.866** | 0.533 | 0.265 | **0.086** | 0.282 |
| w/ RENT (Ours) | 0.859 | **0.548** | **0.339** | 0.082 | **0.332** |
| *Qwen2.5-1.5B-Instruct* | 0.745 | 0.548 | 0.251 | 0.026 | 0.247 |
| w/ Format reward only | **0.754** | 0.558 | 0.259 | 0.054 | **0.271** |
| w/ RENT (Ours) | 0.748 | **0.597** | **0.298** | 0.072 | 0.267 |
| *Qwen2.5-Math-1.5B-Instruct* | 0.852 | 0.744 | 0.452 | 0.092 | 0.244 |
| w/ Format reward only | 0.857 | 0.756 | 0.490 | 0.117 | 0.276 |
| w/ RENT (Ours) | **0.863** | **0.810** | **0.504** | **0.145** | **0.285** |
| *Qwen2.5-7B-Instruct* | 0.906 | 0.762 | 0.423 | 0.110 | 0.311 |
| w/ Format reward only | **0.913** | 0.774 | 0.458 | 0.156 | 0.338 |
| w/ RENT (Ours) | 0.911 | **0.823** | **0.518** | **0.270** | **0.365** |
| *Qwen2.5-Math-7B-Instruct* | 0.956 | 0.834 | 0.495 | 0.143 | 0.225 |
| w/ Format reward only | 0.957 | 0.873 | 0.560 | 0.154 | 0.340 |
| w/ RENT (Ours) | **0.967** | **0.882** | **0.591** | **0.167** | **0.400** |

### 4.5 Qualitative Samples

Table 3 shows a qualitative sample from GSM8K and the Appendix shows a qualitative sample from AIME. These samples verify that the model indeed learns meaningful reasoning skills via entropy minimization. It is not merely learning to format its answer correctly, or otherwise collapsing to other reward-hacking behavior.

### 4.6 Comparison to Concurrent Work

In this section, we compare RENT to concurrent papers which use intrinsic rewards. We evaluate on GSM8K, MATH500, AMC, AIME, and GPQA and run all experiments with Qwen2.5-7B-Instruct as the baseline model. Table 2 shows comparisons to the following methods:

- Test-Time Reinforcement Learning (TTRL) (Zuo et al., 2025) assigns a reward of 1 to the majority answer and 0 to all other answers. In our experiments, we reimplemented this majority voting reward in our codebase.

- Intuitor (Zhao et al., 2025) uses the forward KL divergence between a uniform distribution and the model's distribution as the reward. In contrast, we use entropy, which is the reverse KL divergence from the uniform distribution. Intuitor is mode-seeking while RENT is mode-covering. We ran the publicly available Intuitor code (which is also implemented on top of verl framework Sheng et al. (2024)) with the same batch size, epochs and evaluation strategy as RENT for fair comparison.

- Shao et al. (2025) suggested that even random or "spurious" rewards could be used to improve reasoning. To compare against spurious rewards, we modify our code to set the reward for every generation randomly to 0 or 1 with equal probability. Intuitively, we believe spurious rewards might work because gradients from correct examples contribute to learning, while gradients from incorrect examples might cancel each other out. Our hypothesis is supported by work such as Rolnick et al. (2017), which shows that learning can still happen even when diluting datasets with incorrect labels.

Empirically, we find that RENT is the best of the four methods on average. Compared to TTRL and Intuitor, performance is similar on most benchmarks except AIME, where RENT outperforms both by a

large margin. This is especially interesting since AIME is the hardest benchmark in our evaluations (i.e., the initial accuracy of the model is the lowest). Spurious rewards are not competitive with the other three methods; we conclude that random reward values are not enough and it is indeed beneficial to use meaningful unsupervised rewards that explicitly encourage some measure of confidence. Additionally, we discuss more concurrent works in the Appendix.

Table 2: Comparison of RENT with three concurrent papers: TTRL (Zuo et al., 2025), Intuitor (Zhao et al., 2025), and Spurious Rewards (Shao et al., 2025). The best result on each benchmark is indicated in bold. RENT is the best-performing method on MATH500, AIME, GPQA, and is the best on average.

| | GSM8K | MATH500 | AMC | AIME | GPQA | Average |
|---|---|---|---|---|---|---|
| *Qwen2.5-7B-Instruct* | | | | | | |
| w/ TTRL | **0.933** | 0.822 | 0.521 | 0.172 | 0.346 | 0.559 |
| w/ Intuitor (forward KL) | 0.929 | 0.783 | **0.525** | 0.200 | 0.337 | 0.555 |
| w/ Spurious Rewards | 0.910 | 0.774 | 0.459 | 0.156 | 0.342 | 0.528 |
| w/ RENT (Ours) | 0.911 | **0.823** | 0.518 | **0.270** | **0.365** | **0.577** |

Table 3: The model splits the problem into its key steps and correctly calculates the target age

| **Prompt** | system |
|---|---|
| | Please reason step by step, and put your final answer within \boxed{}. |
| | user |
| | Jaymee is 2 years older than twice the age of Shara. If Shara is 10 years old, how old is Jaymee? Let's think step by step and output the final answer within "\boxed{}". |
| | assistant |
| **Response** | To determine Jaymee's age, we need to follow these steps: |
| | 1. Identify Shara's age. |
| | 2. Calculate twice Shara's age. |
| | 3. Add 2 years to twice Shara's age. |
| | Step 1: Shara's age is given as 10 years old. |
| | Step 2: Twice Shara's age is calculated as: |
| | $$2 \times 10 = 20$$ |
| | Step 3: Jaymee is 2 years older than twice Shara's age, so we add 2 years to 20: |
| | $$20 + 2 = 22$$ |
| | Therefore, Jaymee is $\boxed{22}$ years old. |

### 4.7 Which Tokens to Minimize Entropy Over

We investigated which response tokens are the best to minimize entropy over. To do this, we evaluated the following response token selection strategies:

- **last chunk** - split the response into $k$ chunks, and select all tokens in the last chunk

- **first chunk** - split the response into $k$ chunks, and select all tokens in the first chunk

- **last 10 tokens** - select the last $k$ response tokens

- **random** - select $k$ random tokens from the response

- **after </think>** - select all tokens after the thought process (after the "</think>" tag)

- **after \boxed{** - select all tokens after the boxed answer
- **id_match_last** - select the tokens in the model's final answer
- **id_match_all** - select all tokens tokens in the model's final answer and any previous occurrences
- **after =** - minimize all tokens after the equality symbol ("=")

Figure 4 shows the initial correlation between negative entropy and accuracy for each of these strategies on three datasets. While most token selection strategies do result in a positive correlation between confidence and accuracy, we note that the "last chunk" strategy has a significantly higher correlation compared to the "first chunk" strategy. This suggests that the most important tokens to minimize entropy over are tokens that occur later in the response. Furthermore, based on the low correlation results from the "id_match_last" and "id_match_all" strategies, we find that it is not sufficient to simply minimize the entropy of the final answer tokens; this suggests that, counterintuitively, the token-level confidence of the final answer tokens is not well-calibrated to its true response confidence/accuracy.

## 5 Limitations

Fundamentally, unsupervised learning alone is relatively limited compared to methods which are able to use external supervision for learning. Therefore, it is not surprising that our method is not able to match the performance of methods that have access to the ground-truth answers. It is, of course, a possibility for the model to be confidently wrong (see Appendix A.2 for a qualitative example). Overconfidence is a well-known issue with language models and these calibration errors can cause RENT to fail catastrophically. It could be dangerous to deploy such an unsupervised learning method in the real world without any safeguards. However, we generally find empirically that confidence does correlate with accuracy and the performance does improve by using confidence alone. This indicates that even if the model is overconfident on some answers, it is well-calibrated overall.

## 6 Conclusion

We presented RENT, an unsupervised reinforcement learning method which uses entropy as a reward. In our experiments, we showed that by simply minimizing entropy, we can improve the reasoning performance of language models on GSM8K, MATH500, AMC, AIME, and GPQA. Our reward function is general and can be applied on a wide range of domains. We are excited about the possibility of using entropy minimization and, more broadly, unsupervised RL to improve the capabilities of machine learning models in regimes where external supervision is unavailable.

## 7 Broader Impacts Statement

Due to the unsupervised nature of our method, we cannot guarantee that its outputs would be safe, inoffensive, or correct. However, we believe this broadly applies to many methods and we do not believe any specific societal impacts need to be highlighted here.

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

# A   Appendix

## A.1   Hyperparameters

A full list of hyperparameters can be found in Table 4.

## A.2   More Qualitative Samples

Table 5 shows a qualitative sample from AIME. The model indeed learns meaningful reasoning skills via entropy minimization. It is not merely learning to format its answer correctly, or otherwise collapsing to some other reward-hacking behavior.

Table 6 shows a qualitative example of a confident but incorrect response. The model makes an error in step 4, where it does not realize the 5 hits per player ratio it computed is the average hits per player over 5 games and erroneously multiplies this by 6 directly instead of dividing by 5 first. Despite this, the model appears very confident throughout its thought process. Such examples are to be expected in an unsupervised training paradigm such as RENT, however, as confidently wrong answers and confidently right answers are mathematically indistinguishable under our training objective. While some answers might be pushed to be confidently incorrect, the observed increases in overall performance on these datasets in Figure 2 suggests that RENT pushes more answers to be confidently correct than confidently incorrect at test-time.

Table 4: Hyperparameters.

| Hyperparameter | Value |
|---|---|
| Max prompt length | 1024 |
| Max response length | 3072 |
| Batch size | 64 GSM8K |
| | 500 MATH500 |
| | 80 AMC |
| | 30 AIME |
| | 64 Countdown |
| | 196 GPQA |
| Policy mini batch size | 32 GSM8K |
| | 32 MATH500 |
| | 80 AMC |
| | 30 AIME |
| | 32 Countdown |
| | 32 GPQA |
| Policy micro batch size per GPU | 8 |
| Learning rate | $1 \times 10^{-6}$ |
| Weight decay | 0.01 |
| Learning rate warmup | Constant |
| Optimizer | Adam |
| Temperature | 1.0 for train |
| | 0.8 for validation |
| Top $k$ | -1 |
| Top $p$ | 1 |
| Number of samples per example $n$ | 5 |
| Remove padding | True |
| Use KL loss | True |
| KL loss coefficient | 0.001 |
| Clip ratio | 0.2 |
| Grad clip | 1.0 |

Table 5: The model identifies the geometric series and succinctly soles the problem afterwards

| **Prompt** | system |
| --- | --- |
| | Please reason step by step, and put your final answer within \boxed{}. |
| | user |
| | The infinite product |
| | $$\sqrt[3]{10} \cdot \sqrt[3]{\sqrt[3]{10}} \cdot \sqrt[3]{\sqrt[3]{\sqrt[3]{10}}} \cdots$$ |
| | evaluates to a real number $\sqrt{m}$. What is the value of m? Let's think step by step and output the final answer within \boxed{}. |
| | assistant |
| **Response** | To find the value of the infinite product $\sqrt[3]{10} \cdot \sqrt[3]{\sqrt[3]{10}} \cdot \sqrt[3]{\sqrt[3]{\sqrt[3]{10}}} \cdots$, we can express each term in the product as a power of 10. |
| | The first term is $\sqrt[3]{10} = 10^{1/3}$. The second term is $\sqrt[3]{\sqrt[3]{10}} = \left(10^{1/3}\right)^{1/3} = 10^{1/9}$. The third term is $\sqrt[3]{\sqrt[3]{\sqrt[3]{10}}} = \left(10^{1/9}\right)^{1/3} = 10^{1/27}$. And so on. The infinite product can be written as: |
| | $$10^{1/3} \cdot 10^{1/9} \cdot 10^{1/27} \cdots = 10^{1/3+1/9+1/27+\cdots}.$$ |
| | The exponent is an infinite geometric series with the first term $a = \frac{1}{3}$ and common ratio $r = \frac{1}{3}$. The sum $S$ of an infinite geometric series is given by: |
| | $$S = \frac{a}{1-r} = \frac{\frac{1}{3}}{1 - \frac{1}{3}} = \frac{\frac{1}{3}}{\frac{2}{3}} = \frac{1}{2}.$$ |
| | So the infinite product evaluates to: |
| | $$10^{1/2} = \sqrt{10}.$$ |
| | Thus, the value of $m$ is 10. The final answer is: |
| | $$\boxed{10}$$ |

### A.3  Pass@k Curves

Figure 5, Figure 6, and Figure 7 show the pass@k performance curves for AIME, GSM8K, and MATH500 respectively. We note that a decrease in pass@k performance is expected due to the phenomenon discussed in Section A.2; some confidently wrong answers are amplified during training, and thus a correct answer with lower probability of being sampled is pushed to be even more unlikely as training progresses. In fact, due to the formulation of the pass@k metric, it is highly improbable that pass@k would ever increase throughout training with RENT, as the likely answers amplified by RENT are the ones already most likely to be sampled during pass@k evaluation.

### A.4  Response Length

Figure 8 shows the mean response length of Qwen2.5-7B-Instruct over the course of training on AIME. Overall, the response length increases during training.

Table 6: A qualitative example of a calibration error where the model is confidently wrong. In this example, an error in the thought process is not recovered from and propagates to the final anwer.

| Prompt | system
A baseball team has averaged 15 hits per game over their first 5 games. There are 11 players on the team. Their best player has 25 total hits. If the other players continue their average over the next 6 games, how many hits will each player average across the 6 games in total? Let's think step by step and output the final answer within \boxed{}.
assistant |
|---|---|
| Response | To determine how many hits each player will average across the next 6 games, we need to follow these steps:
1. **Calculate the total number of hits for the first 5 games:**

$$\text{Total hits in first 5 games} = 15\,\text{hits/game} \times 5\,\text{games} = 75\,\text{hits}$$

2. **Calculate the total number of hits by the other players in the first 5 games:**

$$\text{Total hits by other players} = 75\,\text{hits} - 25\,\text{hits} = 50\,\text{hits}$$

3. **Calculate the average number of hits per other player in the first 5 games:**

$$\text{Number of other players} = 11 - 1 = 10\,\text{players}$$
$$\text{Average hits per other player in first 5 games} = \frac{50\,\text{hits}}{10\,\text{players}} = 5\,\text{hits/player}$$

4. **Calculate the total number of hits by each other player over the next 6 games:**

$$\text{Average hits per other player in next 6 games} = 5\,\text{hits/player}$$
$$\text{Total hits by each other player in next 6 games} = 5\,\text{hits/player} \times 6\,\text{games} = 30\,\text{hits/player}$$

Therefore, the number of hits each player will average across the next 6 games is $\boxed{30}$. |
| Ground Truth | 6 |

## A.5 More Concurrent Works

Cheng et al. (2025) introduce an entropy-augmented advantage term in RL to encourage longer, deeper chain-of-thought exploration (addressing exploitation-induced plateaus). This is different from our test-time scenario where we show performance gains with no external rewards.

Cui et al. (2025) study the dynamics of RL training for reasoning and identify the entropy collapse problem where policy entropy drops early, stalling performance. While this may be bad for a longer-term training-time scenario, our goal during test-time training is to maximize the immediate performance gains of the model, so we do not believe this should be a downside for our method.

Wang et al. (2025a) analyze which tokens matter in reasoning by examining token-level entropy during RL with verifiable rewards. They find that only a small minority of high-entropy tokens drive most reasoning improvements, and show that restricting policy updates to those tokens yields large gains (a beyond-80/20 effect). We believe this is complementary work; it shows that certain tokens are more impactful during training, but does not use their entropy directly in any kind of reward. Additionally, the focus of this paper is training-time improvements using ground-truth labels, while we focus on unlabeled test-time improvements.

Zhang et al. (2025) propose EMPO, a fully unsupervised method that (like ours) minimizes a model's predictive entropy to incentivize better reasoning. EMPO uses a semantic confidence reward where they first sample many responses, group them into similar meaning, and penalize the model for the number of meaning clusters it has. This is distinct from our token-level entropy minimization, which does not require multiple samples per update, and instead uses the mean token entropy as a lower-level confidence reward.

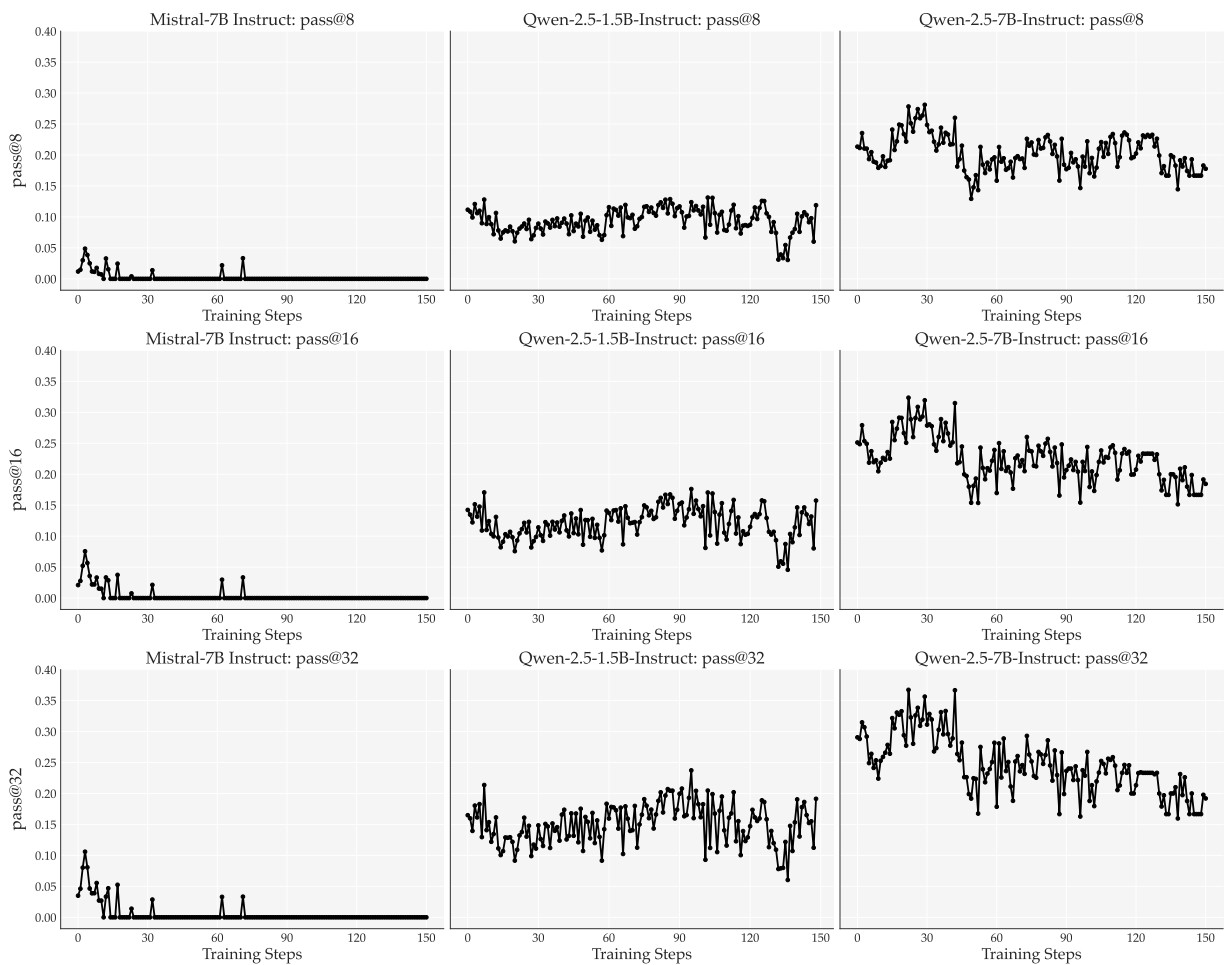

Figure 5: Pass@k curves on the AIME dataset.

Agarwal et al. (2025) similarly incorporates token level entropy minimization as an unsupervised reward for RL. However, we note that this paper is concurrent work to ours. Additionally, we believe our experiments on token minimization patterns offer unique, meaningful insights into the token minimization technique that distinguish our work from this paper.

Jiao et al. (2024) take a different approach by optimizing reasoning via pseudo feedback from frontier LLMs in a preference-based RL framework. In contrast, our method relies solely on the model's own entropy as the reward to improve reasoning.

Wang et al. (2025b) performs reinforcement learning with verifiable rewards on a single question-answer pair, finding that adding an entropy *maximization* term helps to increase generalization accuracy when evaluating on other questions. Since RENT, however, is a test-time method, it is not intended to generalize to new questions not observed during test-time training. Thus, we see this entropy maximization as complementary to our method, where entropy maximization can help during standard RL training and RENT can still be used to minimize entropy at test-time.

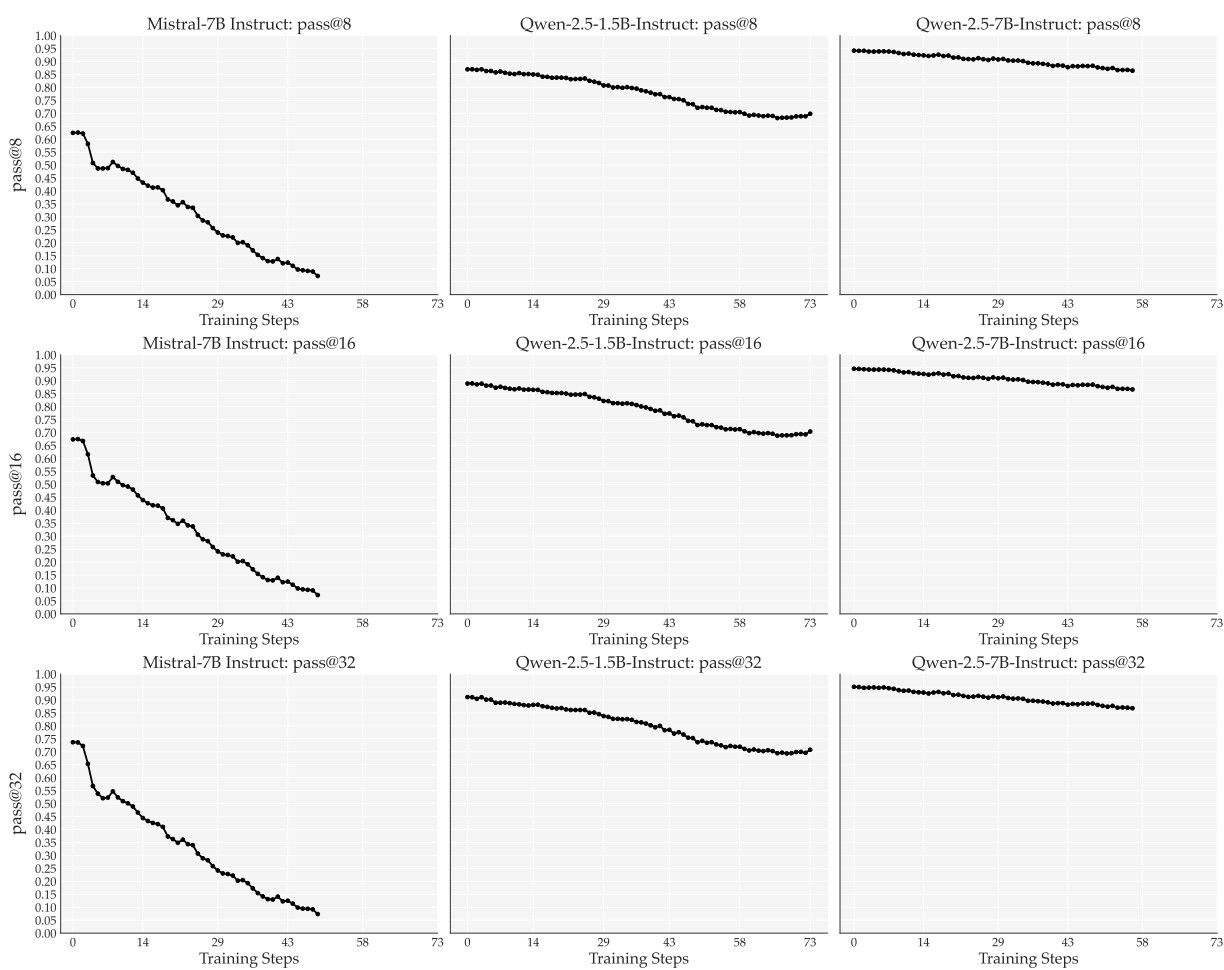

Figure 6: Pass@k curves on the GSM8K dataset.

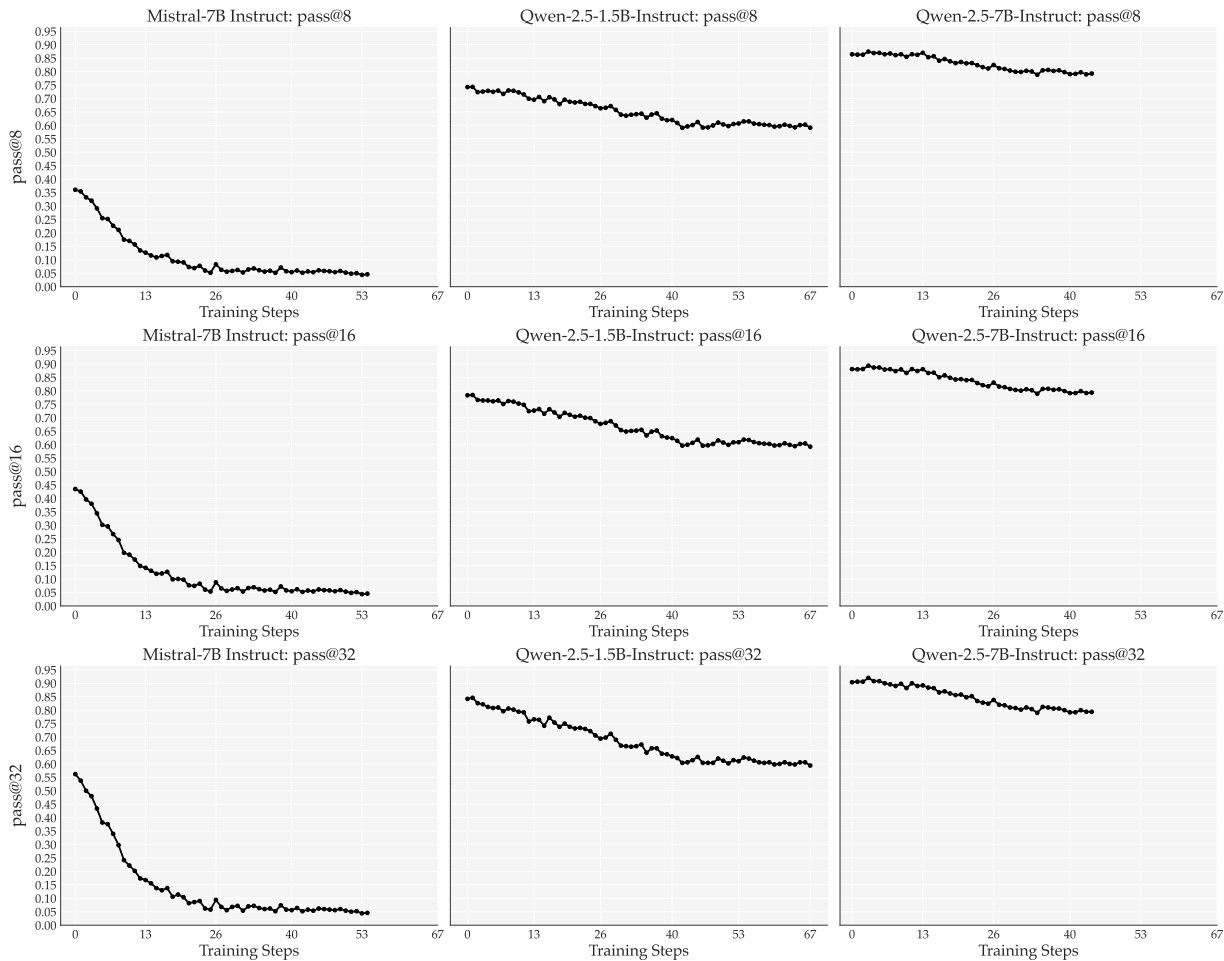

Figure 7: Pass@k curves on the MATH500 dataset.

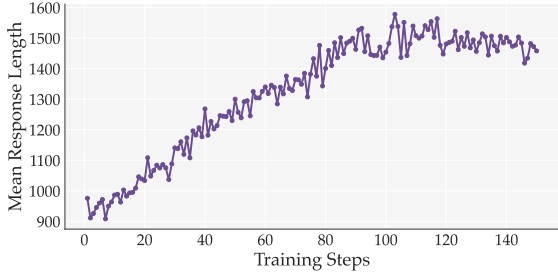

Figure 8: Mean response length of Qwen2.5-7B-Instruct over the course of training on AIME.

