# OpenReview forum: "Maximizing Confidence Alone Improves Reasoning"
_TMLR — Accepted by TMLR_

### Review · Reviewer_t7oV · 2026-02-21

**Summary Of Contributions:**

This paper proposes RENT, a fully unsupervised reinforcement-learning approach for improving LLM reasoning by using the model’s own confidence as the reward, operationalized as the negative average token entropy of the next-token distributions. The authors optimize this intrinsic reward with GRPO and frame the setting as test-time training where the model adapts without access to ground-truth answers. A key empirical contribution is their token-level analysis showing that entropy near the end of the reasoning chain correlates more strongly with accuracy than early tokens, motivating where to apply entropy minimization. They report improvements across multiple reasoning benchmarks (e.g., GSM8K, MATH500, AMC/AIME, GPQA) and several model families and sizes, and they include ablations intended to separate reasoning gains from superficial formatting effects.

**Audience:**

Yes

**Audience Explanation:**

Same as above

**Claims And Evidence:**

Yes

**Claims Explanation:**

I find the paper interesting because it demonstrates a surprisingly strong result: simply pushing the policy toward lower entropy outputs can yield measurable reasoning gains without any external verifier, labels, or reward engineering, and the reward is dense and cheap to compute. The idea is conceptually clean and easy to implement, and the token-selection analysis is particularly helpful because it turns “minimize entropy” from a blunt instrument into something closer to a targeted intervention near the decision point of the solution.

**Requested Changes:**

The weaknesses are also fairly clear and limit how persuasive the “unsupervised” framing is for the main benchmarks used here. For verifiable tasks, RLVR already has labels and straightforward reward construction (e.g., exact match), so it is unclear when one should prefer RENT over verifiable rewards, and the method is fundamentally vulnerable to the well-known failure mode of being confidently wrong. In addition, the work would be much stronger with direct comparisons to RAFT. The formatting study is a good sanity check, but there are cases where the format-only reward matches or even slightly outperforms RENT, and once the format reward saturates (all samples are valid), a binary constant reward can collapse the within-group signal that GRPO relies on, leaving noise and regularizers to dominate updates. Finally, generalization and “general utility” are not convincingly addressed: in many settings the adaptation uses the same dataset for training and evaluation, which raises an overfitting concern, and it remains unclear whether entropy minimization induces entropy collapse behavior that harms broad instruction-following quality. I would also like to see experiments on code-generation benchmarks, where overconfidence can be particularly costly and where utility regression tests tend to be more revealing.

---

> ### Author Response · Authors · 2026-04-20
>
> We thank the reviewer for the thoughtful comments. We address each concern below:
>
> >*For verifiable tasks, RLVR already has labels and straightforward reward construction (e.g., exact match), so it is unclear when one should prefer RENT over verifiable rewards*
>
> We frame our paper as a test-time optimization method. In contrast to RLVR, we show that you can improve performance at test time *without* the use of verifiable rewards (which would leak information to the model at test time). This means you can train with RENT directly on your test set and see a performance improvement.
>
> >*In addition, the work would be much stronger with direct comparisons to RAFT*
>
> We think RAFT is an orthogonal work that may improve test-time results, but in a different way. While RAFT focuses on training models to manage RAG data at test time, our method actively fine-tunes the model at test time to improve performance. These methods could be combined to potentially improve performance even more as well.
>
>
>
> >*the method is fundamentally vulnerable to the well-known failure mode of being confidently wrong*
>
> We acknowledge that this is indeed a failure mode in our limitations section, as the correlation we observed between increased confidence resulting in increased correctness is not guaranteed. (see response to Reviewer iZhZ for examples of being confidently wrong). However, despite no other feedback besides confidence, we show that even with the pitfall of being confidently wrong, models trained with RENT still show performance increases.
>
> >*The formatting study is a good sanity check, but there are cases where the format-only reward matches or even slightly outperforms RENT, and once the format reward saturates (all samples are valid), a binary constant reward can collapse the within-group signal that GRPO relies on, leaving noise and regularizers to dominate updates.*
>
> In addition to comparing to format rewards, we also compare to spurious rewards, which shows that our performance comes from more than just regularization and noise.
>
> >*Finally, generalization and “general utility” are not convincingly addressed: in many settings the adaptation uses the same dataset for training and evaluation, which raises an overfitting concern, and it remains unclear whether entropy minimization induces entropy collapse behavior that harms broad instruction-following quality*
>
> We would first like to clarify that test-time training (TTT) is a very standard setting that is extensively studied in other fields, such as computer vision. We are “training” on the test set, but we do not assume access to the labels. The goal is *not* generalization to another benchmark but rather to directly improve performance on the test set. It is simply another way of using test-time compute to improve performance.
>
> That said, we have added the experiment (see table below) which shows the results of training on AIME24 and testing on MATH500 and GPQA. The model slightly improves on these test sets, but again, we emphasize this is not the point of our paper.
>
> | Benchmark | Base | RENT (trained on AIME) |
> | :--- | :--- | :--- |
> | **Math500** | 0.762 | 0.774 |
> | **GPQA** | 0.312 | 0.358 |
>
> >*I would also like to see experiments on code-generation benchmarks, where overconfidence can be particularly costly and where utility regression tests tend to be more revealing.*
>
>
> We demonstrate our performance increases on the LiveCodeBench benchmark below:
>
> | Model | Base | RENT |
> | :--- | :--- | :--- |
> | **Qwen7B-Instruct** | 0.489 | 0.563 |
> | **Qwen1.5B-Instruct** | 0.125 | 0.229 |

---

### Review · Reviewer_iZhZ · 2026-02-22

**Summary Of Contributions:**

Summary:
This paper introduces RENT, an unsupervised RL framework that utilizes entropy as an intrinsic reward without requiring external labels. Extensive experiments on benchmarks like GSM8K and GPQA across various model families demonstrate that RENT consistently boosts performance on diverse reasoning tasks.

Strengths:
- Methodological Simplicity: The proposed RENT framework is simple, easy to implement, and eliminates the dependency on ground-truth labels or external reward models.
- Broad Empirical Validation: The methodology is supported by comprehensive experiments across diverse model families and reasoning benchmarks, demonstrating generalizability and scaling potential.
- Clarity of Exposition: The paper is well-structured, and the presentation is clear, making it easy to follow.

Weaknesses:
- Dependence on Base Model Quality: As a self-bootstrapping approach, RENT's effectiveness is inherently capped by the base model's initial reasoning capabilities and its calibration between confidence and correctness.
- Marginal Empirical Improvements: While consistent, the performance gains reported in the experiments are relatively modest, raising questions about the method's practical impact compared to other established methods.

**Audience:**

Yes

**Audience Explanation:**

The paper addresses the bottleneck of reward engineering by demonstrating that model entropy can effectively serve as an intrinsic reward for reasoning. This finding is highly relevant to unsupervised learning and the development of self-improving agents. Its simplicity and broad applicability across diverse architectures make it a practical contribution.

**Claims And Evidence:**

Yes

**Claims Explanation:**

The paper’s claims are substantiated by rigorous empirical evaluations across diverse reasoning benchmarks (e.g., GSM8K, MATH500, and GPQA) and multiple model architectures, including Qwen, Mistral, and Llama. The authors provide ablation studies and mechanistic analyses, which offer insights into how entropy minimization drives reasoning improvements. Overall, the consistent performance gains provide convincing evidence for the effectiveness of the proposed unsupervised RL framework.

**Requested Changes:**

1. Impact on Reasoning Diversity: Given that entropy minimization tends to sharpen output distributions, the authors should quantify the impact on reasoning diversity. Providing metrics like pass@k (k > 1) would help clarify whether the performance gains come at the cost of the model's ability to explore alternative solution paths.

2. Analysis of Training Dynamics: The authors could consider providing a more comprehensive analysis of training dynamics and scaling behavior. This could include investigating how the base models' initial capabilities influence effectiveness, evaluating performance stability over an extended number of training steps, and conducting a qualitative error analysis of cases (such as where the model exhibits high confidence in incorrect reasoning).

---

> ### Author Response · Authors · 2026-04-20
>
> We thank the reviewer for the thoughtful comments. We address each concern below:
>
> >*Given that entropy minimization tends to sharpen output distributions, the authors should quantify the impact on reasoning diversity. Providing metrics like pass@k (k > 1) would help clarify whether the performance gains come at the cost of the model's ability to explore alternative solution paths.*
>
> See table below for some pass@k results. It is true that pass@k goes down, but we know that even standard RL can hurt pass@k.
>
> pass@8
>
> |                          | AIME24 |       | MATH500 |       | GSM8K |       |
> |--------------------------|--------|-------|---------|-------|-------|-------|
> |                          |        | +RENT |         | +RENT |       | +RENT |
> | Qwen7B-Instruct          | 0.213  | 0.236 | 0.845   | 0.798 | 0.942 | 0.938 |
> | Qwen-1.5B-Instruct       | 0.112  | 0.106 | 0.743   | 0.595 | 0.869 | 0.850 |
> | Mistral-7B-Instruct-v0.3 | 0.012  | 0.033 | 0.360    | 0.330  | 0.624 | 0.512 |
>
>
> pass@16
>
> |                          | AIME24 |       | MATH500 |       | GSM8K |       |
> |--------------------------|--------|-------|---------|-------|-------|-------|
> |                          | Base   | RENT  | Base    | RENT  | Base  | RENT  |
> | Qwen7B-Instruct          | 0.251  | 0.233 | 0.881   | 0.790  | 0.946 | 0.941 |
> | Qwen-1.5B-Instruct       | 0.142  | 0.129 | 0.784   | 0.605 | 0.889 | 0.865 |
> | Mistral-7B-Instruct-v0.3 | 0.021  | 0.033 | 0.434   | 0.394 | 0.673 | 0.528 |
>
> pass@32
>
>
> |                          | AIME24 |       | MATH500 |       | GSM8K |       |
> |-|-|-|-|-|-|-|
> |                          | Base   | RENT  | Base    | RENT  | Base  | RENT  |
> | Qwen7B-Instruct          | 0.291  | 0.233 | 0.904   | 0.802 | 0.951 | 0.945 |
> | Qwen-1.5B-Instruct       | 0.165  | 0.159 | 0.842   | 0.606 | 0.911 | 0.881 |
> | Mistral-7B-Instruct-v0.3 | 0.035  | 0.033 | 0.562   | 0.500 | 0.737 | 0.547 |
>
> >*The authors could consider providing a more comprehensive analysis of training dynamics and scaling behavior. This could include investigating how the base models' initial capabilities influence effectiveness, evaluating performance stability over an extended number of training steps*
>
> We did include some learning curves in the paper to try to show the training dynamics of the model, and we included experiments across multiple model sizes to try to show the scaling behavior.

---

> > ### Author Response · Authors · 2026-04-20
> >
> > >*Conducting a qualitative error analysis of cases (such as where the model exhibits high confidence in incorrect reasoning).*
> >
> > See below for a qualitative error analysis of the confidently wrong situation:
> >
> > ### Prompt
> >
> > A baseball team has averaged 15 hits per game over their first 5 games. There are 11 players on the team. Their best player has 25 total hits. If the other players continue their average over the next 6 games, how many hits will each player average across the 6 games in total? Let's think step by step and output the final answer within "\boxed{}".
> > assistant
> >
> >
> > ### Model Response
> >
> > To determine how many hits each player will average across the next 6 games, we need to follow these steps:
> > 1. **Calculate the total number of hits for the first 5 games:**
> >    $$\text{Total hits in first 5 games} = \text{Average hits per game} \times \text{Number of games}$$
> >    $$\text{Total hits in first 5 games} = 15 \, \text{hits/game} \times 5 \, \text{games}$$
> >    $$\text{Total hits in first 5 games} = 75 \, \text{hits}$$
> >
> > 2. **Calculate the total number of hits by the other players in the first 5 games:**
> >    $$\text{Total hits by best player} = 25 \, \text{hits}$$
> >    $$\text{Total hits by other players} = \text{Total hits in first 5 games} - \text{Total hits by best player}$$
> >    $$\text{Total hits by other players} = 75 \, \text{hits} - 25 \, \text{hits}$$
> >    $$\text{Total hits by other players} = 50 \, \text{hits}$$
> >
> > 3. **Calculate the average number of hits per other player in the first 5 games:**
> >    $$\text{Number of other players} = \text{Total number of players} - \text{Number of best player}$$
> >    $$\text{Number of other players} = 11 \, \text{players} - 1 \, \text{player}$$
> >    $$\text{Number of other players} = 10 \, \text{players}$$
> >    $$\text{Average hits per other player in first 5 games} = \frac{\text{Total hits by other players}}{\text{Number of other players}}$$
> >    $$\text{Average hits per other player in first 5 games} = \frac{50 \, \text{hits}}{10 \, \text{players}}$$
> >    $$\text{Average hits per other player in first 5 games} = 5 \, \text{hits/player}$$
> >
> > 4. **Calculate the total number of hits by each other player over the next 6 games:**
> >    $$\text{Average hits per other player in next 6 games} = \text{Average hits per other player in first 5 games}$$
> >    $$\text{Average hits per other player in next 6 games} = 5 \, \text{hits/player}$$
> >    $$\text{Total hits by each other player in next 6 games} = \text{Average hits per other player in next 6 games} \times \text{Number of games}$$
> >    $$\text{Total hits by each other player in next 6 games} = 5 \, \text{hits/player} \times 6 \, \text{games}$$
> >    $$\text{Total hits by each other player in next 6 games} = 30 \, \text{hits/player}$$
> >
> > Therefore, the number of hits each player will average across the next 6 games is $\boxed{30}$.
> >
> >
> > ### Analysis
> >
> > Here the model makes an error in step 4, where it does not realize the 5 hits / per player ratio it computed is the average hits / player over 5 games and erroneously multiplies this by 6 directly  instead of dividing by 5 first. Despite this, the model appears very confident throughout its thought process.

---

### Review · Reviewer_TAfA · 2026-04-07

**Summary Of Contributions:**

The paper introduces RENT an unsupervised RL method for LLM reasoning that rewards models to follow high-confidence reasoning paths. The basic idea is just to use the average negative token entropy as reward, i.e. to *minimize* entropy. The paper shows that this works well across various models and benchmarks (GSM8K, MATH500, AMC, AIME, GPQA). By comparing to a format-only reward, they show that the improvements are not just due to learning the right answer format (e.g. using "boxed" environment), which is an important baseline because such effects have been found to explain previous surprising results. They also perform a token-level analysis, showing that it is the later tokens whose entropy most correlates with accuracy -- an interesting and intuitive finding. Finally, they compare to other unsupervised RL methods such as majority voting, forward KL and random rewards, and show that their method outperforms.

**Audience:**

Yes

**Audience Explanation:**

The main finding that minimizing entropy helps is quite surprising, given the long history of entropy maximization as a way to preserve the capacity for exploration. As such I think the paper will certainly be of interest to the community. I am not sure if the method itself will find wide usage given that RLVR still works better, and test-time adaptation can be cumbersome in real-world applications, but it is possible it will find uses and in any case as mentioned it is a very interesting and surprising finding.

**Broader Impact Concerns:**

no concerns

**Claims And Evidence:**

Yes

**Claims Explanation:**

The paper is fairly thorough, showing that the (significant) gains from their method are not due to trivial format learning but reflect reasoning improvements. Overall the ideas and experiments are quite simple, so there is not much to say, but I find them convincing.

**Requested Changes:**

It would be interesting to see if there are other ways to exploit the same phenomenon. For instance, instead of using neg entropy as reward, we could simply sample using a low temperature or top-p value, and then SFT on the resulting sequences. It would be interesting to see how that compares.

One paper that deserves discussion is the following "entropy mechanism" paper, which shows that accuracy can be predicted from policy entropy using a simple functional form. This is highly consistent with the findings of the present paper.

Ganqu Cui, Yuchen Zhang, Jiacheng Chen, Lifan Yuan, Zhi Wang, Yuxin Zuo, Haozhan Li, Yuchen Fan, Huayu Chen, Weize Chen, Zhiyuan Liu, Hao Peng, Lei Bai, Wanli Ouyang, Yu Cheng, Bowen Zhou, and Ning Ding. The Entropy Mechanism of Reinforcement Learning for Reasoning Language Models. arXiv preprint arXiv:2505.22617, 2025.

Another paper shows that RL on a single example works for reasoning, and that in this context entropy maximization is actually very important. Also deserves some discussion.

Yiping Wang, Qing Yang, Zhiyuan Zeng, Liliang Ren, Liyuan Liu, Baolin Peng, Hao Cheng, Xuehai He, Kuan Wang, Jianfeng Gao, Weizhu Chen, Shuohang Wang, Simon Shaolei Du, and Yelong Shen. Reinforcement Learning for Reasoning in Large Language Models with One Training Example. arXiv preprint arXiv:2504.20571, 2025.

---

> ### Author Response · Authors · 2026-04-21
>
> We thank the reviewer for the thoughtful comments. We address each concern below:
> >*It would be interesting to see if there are other ways to exploit the same phenomenon. For instance, instead of using neg entropy as reward, we could simply sample using a low temperature or top-p value, and then SFT on the resulting sequences. It would be interesting to see how that compares.*
>
> In the following experiment, we collect the top-p=1 responses on the GSM8K dataset. Then, we train the model via SFT on its own sampled completions. Afterwards, we evaluate the validation set of GSM8K.
>
> | Model | Performance Prior to SFT | Performance After SFT |
> | :--- | :--- | :--- |
> | Qwen-7B-Instruct | 0.906 | 0.912 |
> | Qwen-Math-7B-Instruct | 0.956 | 0.953 |
>
> Note that this was evaluated in the same manner as our tests with RENT on GSM8K, where we train and test on GSM8K’s train and test subsets respectively.
>
> Additionally, we ran the same experiment with Qwen-7B-Instruct on AIME24 to observe results on a smaller dataset (with the same train and test set)
>
> | Model | Performance Prior to SFT | Performance After SFT |
> | :--- | :--- | :--- |
> | Qwen-7B-Instruct | 0.110 | 0.100 |
>
> We see that pure SFT does not perform as well as RENT, and even degrades performance in some of our experiments.
>
> >*One paper that deserves discussion is the following "entropy mechanism" paper, which shows that accuracy can be predicted from policy entropy using a simple functional form. This is highly consistent with the findings of the present paper.*
>
> We discussed Cui et al. [1] in Appendix A5!
>
> >*Another paper shows that RL on a single example works for reasoning, and that in this context entropy maximization is actually very important. Also deserves some discussion.*
>
> Thanks for the suggestion. We will add a discussion of Wang et al. [2] to the final version!
>
> [1] Ganqu Cui, Yuchen Zhang, Jiacheng Chen, Lifan Yuan, Zhi Wang, Yuxin Zuo, Haozhan Li, Yuchen Fan, Huayu Chen, Weize Chen, Zhiyuan Liu, Hao Peng, Lei Bai, Wanli Ouyang, Yu Cheng, Bowen Zhou, and Ning Ding. The Entropy Mechanism of Reinforcement Learning for Reasoning Language Models. arXiv preprint arXiv:2505.22617, 2025.
> [2] Yiping Wang, Qing Yang, Zhiyuan Zeng, Liliang Ren, Liyuan Liu, Baolin Peng, Hao Cheng, Xuehai He, Kuan Wang, Jianfeng Gao, Weizhu Chen, Shuohang Wang, Simon Shaolei Du, and Yelong Shen. Reinforcement Learning for Reasoning in Large Language Models with One Training Example. arXiv preprint arXiv:2504.20571, 2025.

---

### Decision · Action_Editor_KNAL · 2026-06-08

**Recommendation:** Accept with minor revision

**Additional Comments:**

The paper is sound and the core finding is valuable. Minor revisions should include: (1) explicit discussion of Wang et al.'s single-example RL work and its contrast with entropy maximization; (2) clearer framing of RENT as test-time training rather than generalizable pretraining; (3) brief analysis of failure modes where high confidence correlates with incorrect reasoning, beyond the single qualitative example.

**Audience:**

Yes

**Audience Explanation:**

The result that confidence maximization alone improves reasoning is counter-intuitive and significant for the unsupervised/test-time adaptation community. It challenges the conventional wisdom that exploration requires entropy regularization, offering a minimalist alternative to reward engineering.

**Claims And Evidence:**

Yes

**Claims Explanation:**

The authors demonstrate that entropy minimization serves as a viable intrinsic reward for LLM reasoning improvement without external supervision. The evidence is multi-faceted: (1) consistent gains across GSM8K, MATH500, AMC, AIME, and GPQA with models from Qwen, Mistral, and Llama families; (2) a critical format-reward ablation showing improvements are not merely formatting artifacts; (3) token-level analysis establishing that late-token entropy correlates more strongly with accuracy; (4) comparisons against spurious rewards and majority voting. The authors also adequately addressed reviewer concerns regarding SFT baselines, pass@k diversity trade-offs, and cross-benchmark generalization.